# Modification of Phosphorylation Sites in the Yeast Lysine Methyltransferase Set5 Exerts Influences on the Mitogen-Activated Protein Kinase Hog1 under Prolonged Acetic Acid Stress

Pei-Liang Ye,[a] Bing Yuan,[a] Xue-Qing Wang,[a] Ming-Ming Zhang,[a] Xin-Qing Zhao[a]

[a]State Key Laboratory of Microbial Metabolism, Joint International Research Laboratory of Metabolic & Developmental Sciences, School of Life Sciences and Biotechnology, Shanghai Jiao Tong University, Shanghai, China

**ABSTRACT** Responses to acetic acid toxicity in the budding yeast *Saccharomyces cerevisiae* have widespread implications in the biorefinery of lignocellulosic biomass and food preservation. Our previous studies revealed that Set5, the yeast lysine methyltransferase and histone H4 methyltransferase, was involved in acetic acid stress tolerance. However, it is still mysterious how Set5 functions and interacts with the known stress signaling network. Here, we revealed that elevated phosphorylation of Set5 during acetic acid stress is accompanied by enhanced expression of the mitogen-activated protein kinase (MAPK) Hog1. Further experiments uncovered that the phosphomimetic mutation of Set5 endowed yeast cells with improved growth and fermentation performance and altered transcription of specific stress-responsive genes. Intriguingly, Set5 was found to bind the coding region of *HOG1* and regulate its transcription, along with increased expression and phosphorylation of Hog1. A protein-protein interaction between Set5 and Hog1 was also revealed. In addition, modification of Set5 phosphosites was shown to regulate reactive oxygen species (ROS) accumulation, which is known to affect yeast acetic acid stress tolerance. The findings in this study imply that Set5 may function together with the central kinase Hog1 to coordinate cell growth and metabolism in response to stress.

**IMPORTANCE** Hog1 is the yeast homolog of p38 MAPK in mammals that is conserved across eukaryotes, and it plays crucial roles in stress tolerance, fungal pathogenesis, and disease treatments. Here, we provide evidence that modification of Set5 phosphorylation sites regulates the expression and phosphorylation of Hog1, which expands current knowledge on upstream regulation of the Hog1 stress signaling network. Set5 and its homologous proteins are present in humans and various eukaryotes. The newly identified effects of Set5 phosphorylation site modifications in this study benefit an in-depth understanding of eukaryotic stress signaling, as well as the treatment of human diseases.

**KEYWORDS** *Saccharomyces cerevisiae*, acetic acid stress tolerance, lysine methyltransferase Set5, modification of protein phosphorylation sites, protein kinase Hog1

Address correspondence to Xin-Qing Zhao, xqzhao@sjtu.edu.cn.

The authors declare no conflict of interest.

Acetic acid is commonly present in the hydrolysate of lignocellulosic biomass, it is one of the major toxic substances for the growth and metabolism of the budding yeast *Saccharomyces cerevisiae*, and its toxicity leads to programmed cell death (1). Therefore, the improvement of acetic acid stress tolerance has raised common interest in lignocellulosic biorefinery (2). In addition, studies of *S. cerevisiae* also serve as an important model for food preservation using organic acid (3). Hence, understanding the

molecular mechanisms underlying the response and tolerance of yeast to acetic acid is of great significance for both sustainable bioproduction and the prevention of fungi that are harmful to bioproduction.

Improvement of yeast stress tolerance is highly desirable for industrial applications. Extensive studies have been performed on the selection of robust strains, characterizing key genes related to stress tolerance and process optimization (2, 4). It was found that cell flocculation, a reversible and calcium-dependent cell aggregation (5), affected yeast stress resistance. More specifically, studies in our group and by other researchers revealed that yeast cell flocculation regulates tolerance to ethanol, furfural, and acetic acid (6–8). However, the in-depth molecular mechanism underlying the affected tolerance remains unclear. In our previous studies, multiomics analysis identified key protein kinases that contribute to acetic acid tolerance in the flocculating yeast SPSC01, and engineering yeast strains by overexpressing protein kinases improved fermentation performance (8). It is of great interest to further explore more critical genes related to stress tolerance for metabolic regulation and the engineering of yeast cell factories.

Hog1 is a mitogen-activated protein kinase (MAPK) in *S. cerevisiae*, and it responds to acetic acid through several pathways, including phosphorylating aquaglyceroporin Fps1, regulating catalase gene *CTT1* expression, and avoiding acetic acid-programmed cell death (AA-PCD) (1, 9, 10). We revealed that Hog1 played a crucial role in resisting acetic acid stress in the flocculating yeast strain (8). Also, Hog1 responds to osmotic, heavy metal, cell wall, and oxidative stresses in various fungi (11–14). Plenty of studies have focused on exploring the signaling transduction process of Hog1 upon osmotic stress in which Hog1 is mainly regulated by MAPKK Pbs2 and phosphatase Ptp2 (12). In addition, Hog1 has been reported to be involved in multilayered control of gene expression (13, 14). Nevertheless, mechanisms underlying the upstream regulators for *HOG1* at the transcription level have not been uncovered.

Set5 is a protein lysine methyltransferase in *S. cerevisiae* and was characterized as a histone H4 methyltransferase (15). We found that Set5 overexpression improved yeast tolerance to acetic acid (16), as well as osmotic and heat stresses (17) in different yeast strains. Until now, the Set5-dependent mechanism to combat stresses remains unclear. Set5 comprises an intervening zinc finger domain and a subfamily of the SET domain (18, 19). The zinc finger domain has been reported to recognize specific DNA or RNA sequences and contribute to interactions with other proteins (20), which suggests that Set5 might bind to target sequences and regulate gene expression. Although the removal of the C-terminal region (CTR) of Set5 did not appear to change the Set5-chromatin interaction, localization of Set5, or methyltransferase activity, some decreases in Set5-DNA binding were observed (21). Intriguingly, mutations of phosphosites within the Set5 CTR domain affected the Set5-chromatin interaction and its catalytic activity (21). Set5's activity also functions together with the histone acetyltransferase complex NuA4 and the global chromatin-modifying complexes, COMPASS, to regulate genotoxic stress responses and cell growth (15). Based on the current knowledge, we speculate that posttranslational modification of Set5 might affect its methyltransferase activity, leading to changed expression of target genes under stress. Furthermore, proteins homologous to Set5 are associated with bone morphogenesis, cancer, and various other diseases in humans and other eukaryotes (22–24). Therefore, studies of Set5 may be beneficial for an in-depth understanding of proteins of this family and related disease therapy.

The combined role of Set5 and the histone H3K4 methyltransferase Set1 in regulating gene transcription at the yeast telomeres was reported previously (19), and modification of Set5 phosphorylation sites affects its catalytic activity (21). We are interested in how Set5 phosphorylation exerts control on gene transcription related to yeast stress tolerance and whether Set5 plays a role through any protein kinase-mediated signaling network. In this work, we provide evidence that modification of Set5 phosphorylation sites affects Hog1 transcription, protein expression, and phosphorylation. We also reveal the association of yeast flocculation with Hog1 expression and phosphorylation. To our best

knowledge, our report is the first to demonstrate the functional connection of the central kinase Hog1 with a protein methyltransferase.

## RESULTS

**Modification of Set5 phosphorylation sites affects yeast acetic acid tolerance.** In the comparative phosphoproteomic data, obtained by comparing the flocculating strain SPSC01 and its nonflocculating mutant strain PLY01 under acetic acid stress (8), we found an elevated phosphorylation level of Set5 in SPSC01 relative to that of PLY01 (1.55-, 1.47-, and 1.19-fold for the phosphosites of S458, S461, and S462, respectively). To investigate the effects of these three phosphosites on yeast stress tolerance, we constructed a series of Set5 single-phosphosite mutant strains *in situ*, respectively. Specifically, in the yeast mutants PLY01-SET5-S458A and PLY01-SET5-S458D, the S458 residue of Set5 was replaced by the alanine substitution (A) and phosphomimetic mutation (D), respectively. A total of 6 mutants corresponding to the 3 phosphosites S458, S461, and S462 were constructed by CRISPR/Cas9-mediated genome editing. The tolerance of PLY01-SET5-S458A and PLY01-SET5-S458D to various environmental stresses was assessed via spot assay (see Fig. S1 in the supplemental material). However, there was no observable difference between growth of the strains. Subsequently, more serine and threonine residues within the CTR domain were replaced by alanine or aspartic acid to obtain strains PLY01-SET5-3A, PLY01-SET5-3D, PLY01-SET5-10A, and PLY01-SET5-10D (Fig. 1A). Among them, the mutated sites of PLY01-SET5-3A and PLY01-SET5-3D are S458, S461, and S462, whose phosphorylation levels were changed significantly in our comparative phosphoproteomic data (8), whereas the sites of PLY01-SET5-10A, and PLY01-SET5-10D were selected based on the previous study (21). Considering the indistinguishable growth of spot assay (Fig. S1), the tested concentration of acetic acid was increased to 7.5 g/L. Notably, the phosphomimetic mutations of Set5 led to better growth and ethanol production efficiency than both the parent strain PLY01 and the corresponding alanine substitution mutant strains under acetic acid stress, despite similar performance under the stress-free condition (Fig. 1B to G and Fig. S2A and B). These results suggest that phosphorylation of Set5 may be involved in response to acetic acid stress in budding yeast cells. However, in terms of fermentation, there is no obvious dosage effect of the phosphorylation sites since PLY01-SET5-10D did not show better performance than PLY01-SET5-S458D and PLY01-SET5-3D (Fig. S2C and D). Additionally, growth under other stresses, including $H_2O_2$, furfural, and hyperosmolarity, was also assessed in a microtiter plate. Except for hyperosmolarity, modification of the Set5 phosphorylation sites led to changes in yeast cell responses to acetic acid, $H_2O_2$, and furfural stress (Fig. S3). Different growth profiles were observed for the different stresses, suggesting stress-specific regulation by the mutant Set5 proteins.

**Set5 alters the transcriptional level of specific acetic acid tolerance-related genes.** Set5 has previously been reported to cooperate with Set1 to repress the expression of genes at retrotransposons and telomeres (25). However, no studies have been reported on the regulation of gene transcription by Set5 under acetic acid stress conditions. We supposed that Set5 may regulate acetic acid tolerance by promoting or repressing the expression of specific genes. Cell samples of strains were collected at the exponential phase in the fermentation with 7.5 g/L acetic acid. The transcriptional levels of 31 acetic acid tolerance-related or potentially regulated genes selected from our comparative omics data (8) were measured (Fig. S4). Above all, the expression of *GPX1* encoding the phospholipid hydroperoxide glutathione peroxidase, and *RCK1* and *SCH9* encoding protein kinases, was obviously affected by the modification of Set5 phosphorylation sites, where *GPX1* and *RCK1* were upregulated by 1.34-fold and 1.23-fold, respectively, and *SCH9* was downregulated by 0.43-fold (Fig. S4A). Previous studies have confirmed that overexpression of *GPX1* or *RCK1* enhanced tolerance to acetic acid (8, 26), and our current results further reveal the association of these two genes with the function of Set5. Deficiency of Sch9, a downstream kinase of target of rapamycin complex 1 (TORC1), switched yeast cells to utilize acetic acid as a carbon source, leading to enhanced accumulation of the stress protectant trehalose (27). The decreased expression

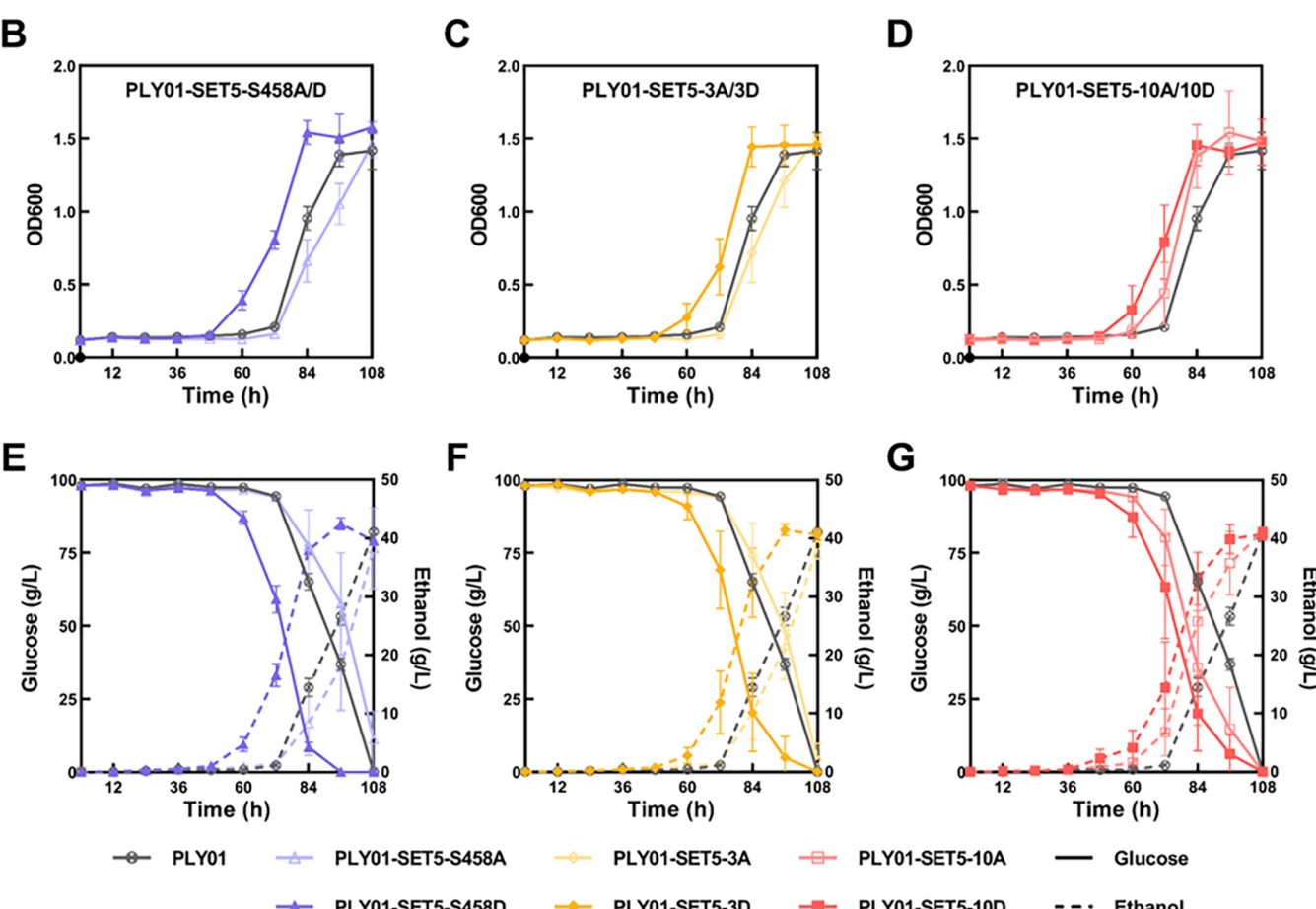

**FIG 1** Modification of Set5 phosphorylation sites affects yeast tolerance to acetic acid. (A) Sequence alignment of Set5 in *S. cerevisiae* PLY01 and various Set5 phosphosite mutant strains. Phosphosites are highlighted in black and bold in Set5-PLY01. Mutated sites are highlighted in red and bold. (B to G) Growth and fermentation performance of PLY01 and Set5 mutant strains under 7.5 g/L acetic acid stress are shown, respectively. (B and E) Comparison between PLY01, PLY01-SET5-S458A, and PLY01-SET5-S458D. (C and F) Comparison between PLY01, PLY01-SET5-3A, and PLY01-SET5-3D. (D and G) Comparison between PLY01, PLY01-SET5-10A, and PLY01-SET5-10D. Data are averages from at least three duplicate experiments. Error bars indicate standard deviations.

of *SCH9* in the mutant strains implied the intervention of Set5 with the stress signaling pathway. Moreover, through comparison between the alanine substitution and phosphomimetic mutation of the same sites, the delta (9) fatty acid desaturase gene *OLE1*, whose overexpression improved tolerance to multiple stresses, including acetic acid (28, 29), was proven to be affected by modification of Set5 phosphorylation sites (Fig. S3D).

Tremendous transcriptome shifts have been reported to occur in a very short period of time (20 min) when facing acetic acid stress in *S. cerevisiae* (30). Therefore, the transcription

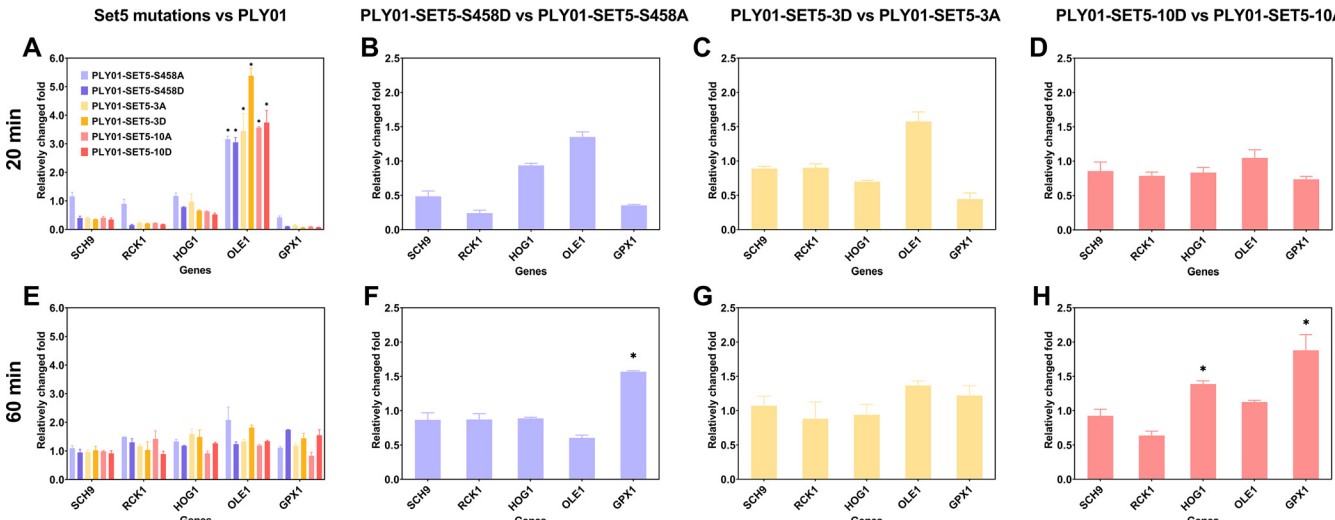

**FIG 2** Transcription of acetic acid tolerance-related genes affected by modification of Set5 phosphorylation sites. (A to D) Cells were collected after 20 min of 7.5 g/L acetic acid treatment. (E to H) Cells were collected after 60 min of 7.5 g/L acetic acid treatment. Comparisons between PLY01 and Set5 mutant strains, or corresponding phosphomimetic and alanine substitution Set5 mutant strains, are shown, respectively. *, $P < 0.05$. Data are averages from at least three duplicate experiments. Error bars indicate standard deviations.

of 5 key genes among the aforementioned 31 genes was measured after 20 or 60 min treatment with 7.5 g/L acetic acid (Fig. 2). Among the genes, only the expression of *OLE1* was significantly upregulated in all of the Set5 mutant strains compared to PLY01 at 20 min (Fig. 2A), no matter whether alanine or aspartic acid was incorporated in the mutations. When the treatment time was increased to 60 min, the expression of *OLE1* dropped back to the normal level, meaning the end of rapid response (Fig. 2E). Besides *OLE1*, the transcriptional level of *HOG1* and *GPX1* was enhanced in PLY01-SET5-10D compared to PLY01-SET5-10A (Fig. 2H).

**Modification of the phosphorylation sites affects the enrichment of Set5 to *HOG1*.** Our recent studies revealed that yeast cell flocculation improved acetic acid stress tolerance in *S. cerevisiae*, and multiomics analyses implied that the protein kinase signaling network contributes to flocculation-associated protection mechanism. We found increased Hog1 expression by yeast flocculation in the proteomic data (8), and we also observed elevated Set5 phosphorylation in the flocculating yeast SPSC01 compared to its nonflocculating mutant PLY01 (8). Intriguingly, our unpublished proteomic data revealed increased expression of Hog1 by overexpressing *SET5* in *S. cerevisiae* BY4741 under acetic acid stress, which led us to investigate the possible functional connection between Set5 and Hog1. In addition, according to the data of chromatin immunoprecipitation (ChIP) sequencing using strains BY4741 and BY4741-SET5ΔZF (deleting zinc finger domain of *SET5* in BY4741), differential Set5 enrichment in both the protein kinase genes of *HOG1* and *RIM15* was found (Fig. S5), and Set5 was enriched in the *HOG1* coding region (Fig. S6). We therefore assumed that phosphorylation of Set5 might regulate the expression of *HOG1* through binding to its gene sequence.

We used different acetic acid concentrations in the studies. When comparing SPSC01 and PLY01, 5 g/L acetic acid was used, whereas when comparing the different PLY01 mutants, we used 7.5 g/L acetic acid because under only this higher concentration we can observe the clear difference in the fermentation performance. We then measured the enrichment of Set5 to several possible target genes in PLY01 under 7.5 g/L acetic acid stress through ChIP assay using the Set5-specific antibody. These genes were selected based on the genome-wide studies of Set5 target genes using the wild-type lab yeast strain *S. cerevisiae* BY4741 and the mutant strain carrying a deletion of the zinc finger domain of Set5 employing ChIP-seq analysis (data not shown). Significant enrichment of Set5 to the *HOG1* sequence was identified in PLY01 (Fig. 3A), similar to the results we observed in *S. cerevisiae* BY4741 (Fig. S6).

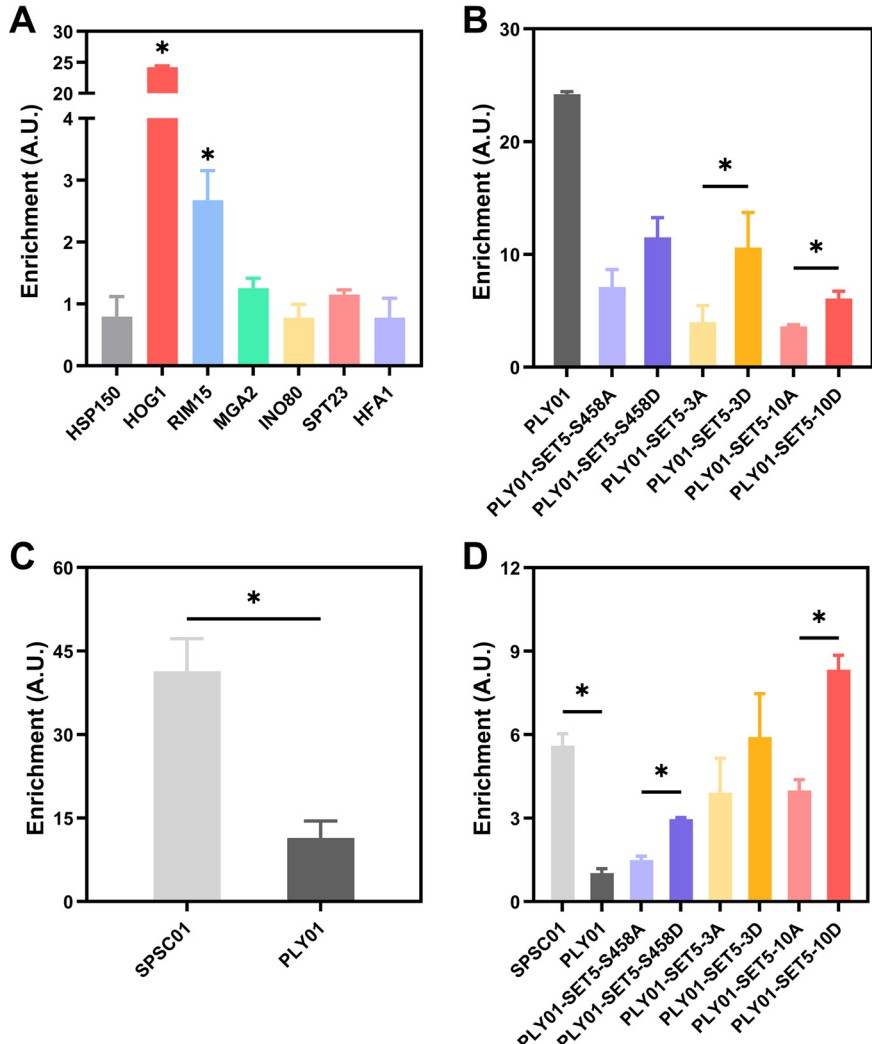

**FIG 3** Set5 is enriched for specific regions of *HOG1*. (A) ChIP-qPCR of Set5 with various selected genes in strain PLY01 under 7.5 g/L acetic acid stress. (B) ChIP-qPCR of Set5 with *HOG1* in PLY01 and Set5 mutant strains under 7.5 g/L acetic acid stress. (C) ChIP-qPCR of Set5 with *HOG1* in SPSC01 and PLY01 under 5.0 g/L acetic acid stress. (D) ChIP-qPCR of Set5 with *HOG1* in SPSC01, PLY01, and the Set5 phosphosites mutant strains under stress-free conditions. *, $P < 0.05$. Data are averages from at least three duplicate experiments. Error bars indicate standard deviations. A.U., arbitrary units.

Subsequently, the effect of Set5 phosphosite mutations on the enrichment in *HOG1* was compared (Fig. 3B). All the Set5 mutant strains showed decreased enrichment in *HOG1* when compared with that of the wild-type strain, indicating that any manipulation of Set5 phosphosites affected its binding to DNA. Among the different mutants, the phosphomimetic mutant strains (PLY01-SET5-S458D, PLY01-SET5-3D, and PLY01-SET5-10D) showed higher enrichment than the corresponding alanine substitution mutant strains (PLY01-SET5-S458A, PLY01-SET5-3A, and PLY01-SET5-10A). We also compared the differences in Set5 enrichment of *HOG1* between the flocculating strain SPSC01 and its nonflocculating mutant strain PLY01 in the presence of 5.0 g/L acetic acid. The results showed that cell flocculation remarkably improved the enrichment of Set5 in *HOG1* (Fig. 3C). Further tests without stress were also performed, and enrichment of Set5 in *HOG1* was observed in all the strains we tested. Generally, under both the stress condition and the stress-free condition, phosphomimetic mutant strains showed higher enrichment than that of the alanine substitution mutant strains, and higher enrichment was observed under the acetic acid stress condition, suggesting promotion by acetic acid. All Set5 mutants showed

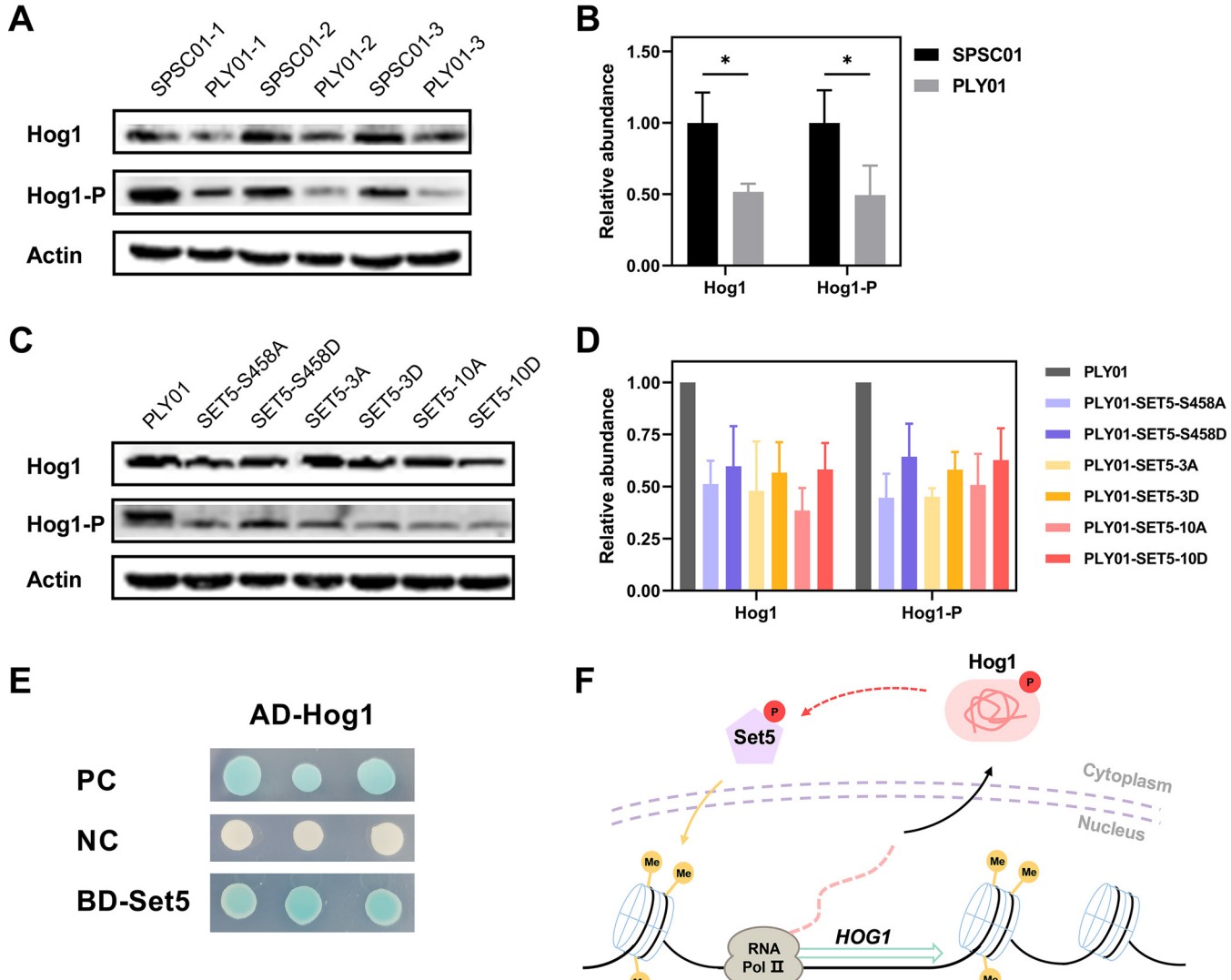

**FIG 4** Hog1 expression and phosphorylation are affected by Set5 and flocculation. (A and B) Abundance of Hog1 and Hog1-P in SPSC01 and PLY01 with three replications under 5.0 g/L acetic acid stress. (C and D) Abundance of Hog1 and Hog1-P in PLY01 and Set5 mutant strains under 7.5 g/L acetic acid stress. Quantitative data of Western blot analysis in panels A and C are shown in panels B and D, respectively. (E) Yeast two-hybrid system confirms that Hog1 interacts with Set5 (blue dots). AD, *GAL4* activation domain; BD, *GAL4* DNA binding domain. Hog1 and Set5 were fused to AD and BD, respectively. PC, positive control; NC, negative control. (F) Schematic diagram of regulated Hog1 and Hog1-P by Set5. *, $P < 0.05$. Data are averages from at least three duplicate experiments. Error bars indicate standard deviations.

a reduction in enrichment compared with PLY01 when the strains were challenged by acetic acid, whereas the opposite trend was seen under no-stress conditions (Fig. 3D).

**Regulation of Hog1 expression and phosphorylation by modification of Set5 phosphorylation sites.** Posttranscriptional regulation mechanisms are widely used for managing protein levels in yeast cells, which leads to inconsistencies between transcriptional levels and protein levels (31). The response of protein kinase Hog1 to acetic acid is phosphorylation dependent (10, 14, 32). We thereby monitored the protein abundance and phosphorylation level of Hog1 using the corresponding antibodies (Table S4). Under acetic acid stress, significantly higher levels of Hog1 and Hog1-P (phosphorylated Hog1) were observed in SPSC01 than in PLY01 (Fig. 4A and B), whereas the relative ratio of Hog1-P to Hog1 was identical. This result is in line with the expectation that improved Hog1 expression led to its higher phosphorylation. The levels of Hog1 and Hog1-P of PLY01 and Set5 phosphosite mutant strains were also measured. Although all mutations caused a decreased abundance of Hog1 or Hog1-P

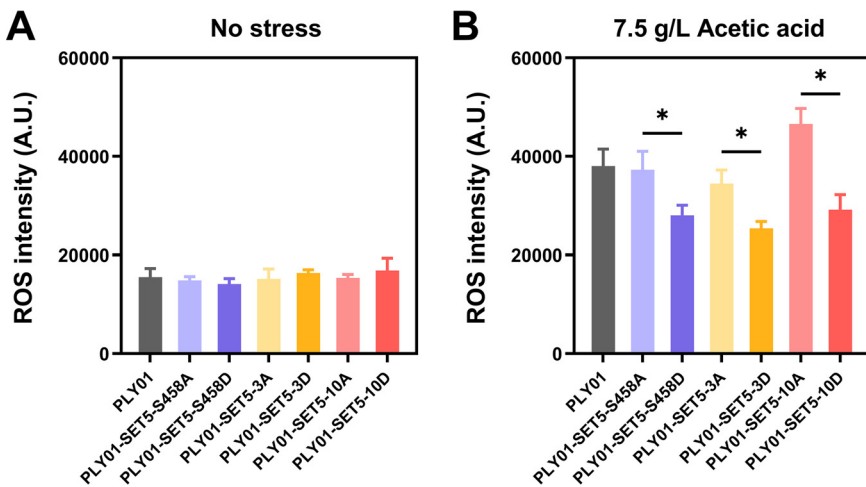

**FIG 5** Modification of Set5 phosphorylation sites affects ROS accumulation. The intracellular ROS accumulation of strain PLY01 and Set5 mutations was tested under no stress (A) or 7.5 g/L acetic acid (B). *, $P < 0.05$. Data are averages from at least three duplicate experiments. Error bars indicate standard deviations.

compared to PLY01, phosphomimetic mutant strains exhibited higher levels of Hog1 and Hog1-P than the alanine substitution mutant strains (Fig. 4C and D). The higher level of phosphorylated Hog1 of PLY01 in the presence of 7.5 g/L acetic acid than with 5.0 g/L acetic acid suggests that more severe acetic acid stress drives higher levels of Hog1 phosphorylation (Fig. 4A versus Fig. 4C).

In comparison to other lysine methyltransferases in *S. cerevisiae*, Set5 is one of the most highly phosphorylated SET domain-containing proteins and a common substrate for numerous kinases (33). Therefore, a yeast two-hybrid assay was performed, and we confirmed the protein-protein interaction between Set5 and Hog1 (Fig. 4E), suggesting that Hog1 might phosphorylate Set5. Due to the challenge in synthesizing a specific phosphorylated Set5 antibody, this hypothesis has not been verified. We assume that there is a positive feedback loop to regulate Hog1 by Set5 in response to acetic acid: phosphorylated Set5 enriches to the *HOG1* sequence and enhances histone H4 methylation in this region; the transcriptional level of *HOG1* is promoted, followed by elevated protein abundance and phosphorylation level; and the activated Hog1 phosphorylates Set5 and facilitates its enrichment and catalysis in turn (Fig. 4F). In the previous study, we examined the effects of multiple protein kinases on yeast stress tolerance (8). Encouraged by the confirmation of Set5 and Hog1 interaction, we also detected interactions between Set5 and other five selected protein kinases whose phosphorylation level was upregulated by flocculation and overexpression, which improved acetic acid tolerance in the previous study (11). The results revealed that besides Hog1, Akl1 and Rim15 also interacted with Set5 (Fig. S7).

**Altered ROS accumulation by modification of Set5 phosphorylation.** Increased reactive oxygen species (ROS) accumulation by acetic acid stress has been reported in previous studies (34). In various genetically manipulated strains, improved tolerance to acetic acid was often accompanied by a decrease in ROS accumulation (8, 35, 36). Therefore, the intracellular ROS accumulation of PLY01 and Set5 mutant strains was measured. All strains maintained the fundamental level of ROS accumulation under the stress-free condition (Fig. 5A). In the presence of 7.5 g/L acetic acid, the Set5 phosphomimetic mutant strains PLY01-SET5-S458D, PLY01-SET5-3D, and PLY01-SET5-10D with better fermentation performance (Fig. 1D and E) showed significantly decreased accumulation of ROS compared to the Set5 alanine substitution mutant strains PLY01-SET5-S458A, PLY01-SET5-3A, and PLY01-SET5-10A (Fig. 5B), suggesting higher cell viability and metabolic activity.

## DISCUSSION

Set5 has been identified as a yeast histone H4 methyltransferase (15) which functions in epigenetic regulation. In addition, the nonchromatin substrates of Set5 were also implicated (21). We report here that the lysin protein methyl transferase Set5 was enriched in the coding region of HOG1 in S. cerevisiae (Fig. 3). It will be meaningful to further investigate whether the binding of Set5 with HOG1 is direct or indirect. We also revealed that modification of Set5 phosphorylation sites affects Hog1 phosphorylation (Fig. 4). These results unveil, for the first time, the connection of Set5 and Hog1 in regulating yeast stress tolerance. To our best knowledge, this is the first report on the functional connection of the methyl transferase Set5 with the central protein kinase Hog1, which deserves further investigation.

Studying the transcription of the selective genes by modification of Set5 phosphorylation sites suggest that substitution of serine or threonine residues in Set5 may promote the repressive effects of Set5 on gene transcription and that phosphorylation at the sites investigated in this study might not be solely responsible for the regulatory effects. Regulation of gene expression is normally controlled by a complicated network. We thus also hypothesize that mutations of the Set5 phosphorylation sites (alanine or aspartic acid) may cause fine-tuned regulation of the downstream genes. For regulation of gene expression, either too much or too little phosphorylation of Set5 may not be beneficial, so the results caused by phosphomimetic and alanine substitution of Set5 may not be completely opposite. Our results are the first to compare the effects of single, triple, and 10 phosphorylation sites of Set5 on gene transcription. Further investigation will reveal how the phosphorylation of Set5 controls gene expression under stress together with multiple regulatory events. In the current results, only several genes were affected by the modification of Set5 phosphorylation sites. However, these results help us better understand the effects of Set5 phosphosite modification and promote further investigation of the effect of Set5 phosphorylation by evaluating more genes, more time points, and more conditions.

Using the ChIP assays, we detected that all the Set5 mutants have shown a reduction in their interaction with HOG1 compared with that of the parent strain PLY01 under acetic acid stress (Fig. 3B). Interestingly, the opposite profile was observed under no-stress conditions (Fig. 3D). We are not clear why such profiles were observed, which warrant further investigation. It is evident that more enrichment of Set5 appears under the stress condition compared with the nonstressful condition (Fig. 3B and D), suggesting that the enrichment of Set5 is promoted by the stress and is responsible for stress signaling. The results also implied that there may exist an additional Set5 partner which promotes the binding of Set5 to HOG1 during acetic acid stress. These results also suggest dynamic regulation of the central protein kinase Hog1 under different environmental conditions. The phosphomimetic and alanine substitution mutations are widely used to study protein phosphorylation (37, 38). However, no studies have been performed to test whether such modifications affect protein folding, stability, or expression levels. It is worthwhile to examine whether the modification of Set5 phosphorylation sites affects the protein levels of the Set5-derivative proteins.

In the previous report, the mutation of Set5 phosphosites caused a change in methyltransferase activity, which affected the methylated modification of histone H4 in the target region (21). Histone methylation has been reported to contribute to the regulation of gene transcription through multiple pathways, including affecting histone chaperone complexes Spt16-Pob3, transcription-replication conflicts (TRCs), the employment of RNA polymerase II, etc. (39, 40). We found that the abundance of Hog1/Hog1-P and ROS accumulation were regulated by individual Set5 phosphomimetic mutation (Fig. 4 and 5). Our results provide clues for further exploration of Set5 phosphorylation in the kinase-mediated stress signaling network. The substitutions we employed are commonly used to study protein phosphorylation (37, 38). However, we cannot rule out the possibility that the mutations may affect protein folding and/or stability of Set5, which deserves further investigation.

We found that under acetic acid stress, decreased ROS accumulation was obvious when the phosphorylation sites were mutated to aspartic acid compared to that of the

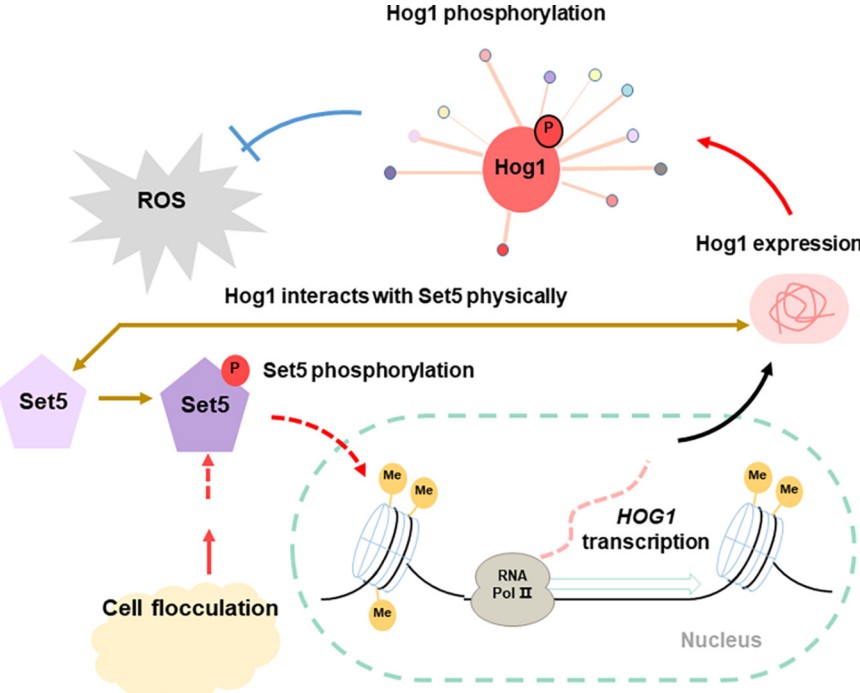

**FIG 6** Schematic diagram of improved acetic acid tolerance by cell flocculation through Set5 and Hog1. Under acetic acid stress, the higher phosphorylation level of Set5 induced by flocculation specifically promotes the expression of *HOG1*. Subsequently, increased Hog1/Hog1-P activates the global kinase network and decreases ROS accumulation.

alanine substitution (Fig. 5B). We propose a possible mechanism underlying lower ROS accumulation by the Set5 aspartate substitutions, where (i) aspartate substitutions mimic phosphorylation of Set5, (ii) phosphorylated Set5 enriches to the *HOG1* sequence and increases *HOG1* transcription, (iii) more *HOG1* mRNA causes an enhanced abundance of Hog1 and Hog-P, (iv) the activated Hog1 regulates Gpx1 expression and activity as revealed in our previous studies (11), and (v) improved Gpx1 activity leads to lower ROS accumulation. However, we cannot rule out that other factors may also contribute to ROS scavenging, which deserves further investigation.

It should be noted that like other lysine methyltransferases, Set5 not only functions as a histone modification enzyme but also has the potential to methylate other nonhistone proteins (21). Future *in vivo* investigation will seek to identify which phosphosite of Set5 is crucial to regulating its function and how Set5 phosphorylation exerts control in stress signaling and expression of important functional proteins.

Based on the results of this study, we propose a possible mechanism underlying improved acetic acid tolerance by yeast cell flocculation, where (i) flocculation facilitates phosphorylation of Set5 via protein kinases (Hog1, Akl1, or Rim15), (ii) phosphorylated Set5 enriches the *HOG1* sequence and methylates histone H4 of this region, (iii) the modification of histone H4 improves *HOG1* transcription, (iv) more mRNA causes enhanced abundance of Hog1/Hog-P, and (v) activated Hog1 acts as the core regulator and works with other protein kinases to efficiently scavenge ROS induced by acetic acid (Fig. 6). The crucial role of Hog1 with other protein kinases and scavenging of ROS by glutathione peroxidase (GPX) have been verified in our previous study (11), and the results in this study provide more clues on the upstream regulation events.

Both Set5 and Hog1 are conserved across eukaryotes, and Hog1 is the yeast homolog of p38 MAPK in mammals. Functions of the homologous proteins of Set5 and Hog1 are associated with the regulation of gene expression, stress signaling, disease occurrence, and development (12, 41, 42). Hog1 has been reported to respond to multiple stresses and be involved in immune evasion in fungal pathogens (13, 41). p38 controls a plethora

of functions, and dysregulation of p38 has been linked to diseases such as immune disorders, inflammation, and cancer (42). Considering the conserved and important roles of Hog1 in fungi and humans, the work in this study benefits a deeper understanding of the regulatory mechanism of Hog1 and discovery of strategies to combat fungal infections and related diseases. Set5 belongs to the SMYD family lysine methyltransferase (21). Our study also contributes to understanding phosphorylation modification of SYMD proteins, which might similarly regulate special protein kinases in various organisms. For example, SMYD2 methylates retinoblastoma tumor suppressor protein (RB), which regulates the cell cycle and is mutated in several kinds of cancer (43). Further studies on the phosphorylation of SYMD proteins may benefit the development of potential therapeutic targets in human cancer.

## MATERIALS AND METHODS

**Strains, media, and materials.** The microbial plasmids and strains used in this study are listed in Tables S1 and S2 in the supplemental material, respectively. *Escherichia coli* DH5$\alpha$ was used for the propagation of plasmids and cultivated in Luria-Bertani medium (5 g/L yeast extract, 10 g/L tryptone, and 10 g/L NaCl). The antibiotics 50 $\mu$g/mL kanamycin or 100 $\mu$g/mL ampicillin were added into LB medium for the selection of transformants.

*S. cerevisiae* strains SPSC01, PLY01 (SPSC01-FLO1$\Delta$), and Set5 mutant strains were cultured in YPD medium (10 g/L yeast extract, 20 g/L peptone, and 20 g/L glucose). We added 300 $\mu$g/mL G418 and 500 $\mu$g/mL hygromycin B into YPD medium for the selection of Set5 mutation transformants. For yeast two-hybrid system strains, SD medium (6.7 g/L yeast nitrogen base without amino acid, 20 g/L glucose, and 0.6 g/L DO supplement [His, Leu, Trp, and Ura]) with 20 mg/L uracil and 20 mg/L histidine was used for the selection of transformants and subsequent protein-protein interaction validation. When performing fermentation experiments, the fermentation medium (100 g/L glucose, 4 g/L yeast extract, and 3 g/L peptone) was used, into which 5.0 or 7.5 g/L acetic acid was supplemented for the stress tolerance test. The strain PLY01 and yeast two-hybrid system strain AH109 were used as the host strains for genetic manipulation.

**Construction of Set5 mutant strains.** The nonflocculating strain was constructed by deleting the *FLO1* gene in the flocculating *S. cerevisiae* SPSC01, termed PLY01 (44). Set5 mutant strains were acquired using PLY01 as the host strain. The construction method of all engineered strains adopted the CRISPR-Cas9 system and lithium acetate transformation (45). The donor cassettes for substituting the *SET5* target sequence were obtained by PCR amplifying from the synthesized fragment. The primers for donor DNA amplification and gRNA construction are listed in Table S3. The Cas9-G418 plasmid was first introduced into PLY01 to obtain strain PLY01-Cas9. The gRNA-SET5S458 plasmid and donor DNA corresponding to different phosphorylation mutations were transformed into PLY01-Cas9 simultaneously. The transformants were verified by diagnostic PCR using the verification primers V-SET5-F (GGTACAACTGTACATACTGAAGAG) and V-SET5-R (CGCTCTCAAATTCACTCTCA). The correctly verified transformants were inoculated in fresh YPD medium without antibiotic and were subcultivated constantly for 3 to 5 days to cure the two plasmids.

**Spot assay.** Spot assay was used for stress tolerance evaluation. After overnight culture, the absorbance (optical density at 600 nm [OD$_{600}$]) of seed was adjusted to ~1.0 with distilled water. Tenfold serially diluted suspensions of seed were spotted on the YPD plates containing a specific concentration of inhibitors, including 10% ethanol, 3.5 mM H$_2$O$_2$, 5.0 g/L acetic acid, 1.0 M NaCl, 0.75 g/L furfural, 40 mM propanoic acid, 2.0 M sorbitol, and 1.0 g/L Congo red. Except for high-temperature stress (41℃), all strains were cultivated at 30℃. The plates were photographed after the appropriate culture.

**Growth and ethanol fermentation evaluation.** For fast and efficient growth evaluation, Bioscreen C (Bioscreen, Finland) was used. When strains grew to exponential phase after overnight cultivation, cells were collected and inoculated into fermentation medium with an initial OD$_{600}$ of 0.1 in a microtiter plate. The growth (OD$_{600}$) of strains was measured and recorded by Bioscreen C within an interval of 0.5 h at 30℃ and medium-speed shaking. We added 5.0 g/L acetic acid, 10 mM H$_2$O$_2$, 5.0 g/L furfural, or 1.0 M NaCl into the medium to evaluate stress tolerance.

Batch fermentation evaluation was performed in a 250-mL flask containing 100 mL fermentation medium with or without acetic acid stress. The inoculation method was consistent with the processes mentioned before. In contrast, the cells of flocculating strain SPSC01 had to be deflocculated and resuspended by 0.1 M sodium citrate buffer (pH 4.5) (46). The fermentation conditions were controlled at 30℃ and 150 rpm without pH adjustment. For analysis of fermentation performance, samples were collected every 12 h until glucose was consumed completely. The concentrations of glucose and ethanol in fermentation broth at each time point were detected via a high-performance liquid chromatography (HPLC) system (Waters Alliance e2695 HPLC; Waters, USA) with a Bio-Rad Aminex HPX-87H column. The column was eluted with 4 mM sulfuric acid at a 0.6-mL/min flow rate at 50℃.

**ROS accumulation.** The ROS accumulation of strains was detected by reactive oxygen species assay kit (Beyotime Biotechnology, China) via supplementation of fluorescent probe 2′,7′-dichlorofluorescein diacetate (DCFH-DA). Cells were harvested at the corresponding exponential phase under no stress (4 h) or 7.5 g/L acetic acid (72 h or 84 h).

**RT-qPCR analysis.** Cell samples were collected at an initial 20 or 60 min after acetic acid treatment or exponential phase in fermentation (72 h or 84 h). Total RNA was extracted and reversely transcribed into cDNA using HiPure yeast RNA kit (Magen, China) and Goldenstar RT6 cDNA synthesis kit (Tsingke, China), respectively, followed by reverse transcriptase quantitative PCR (RT-qPCR) analysis using 2× T5

Fast qPCR mix SYBR green I (Tsingke, China). The mRNA transcriptional levels were normalized using *ALG9* as a reference gene and calculated by the threshold cycle ($2^{-\Delta\Delta CT}$) method (39). The primers used for RT-qPCR analysis are listed in Table S3.

**Chromatin immunoprecipitation assay.** ChIP was performed as previously described with a few modifications (40). Briefly, cells were grown to exponential phase, cross-linked in 1% formaldehyde, and subsequently quenched cross-linking in 0.125 M glycine. Yeast cells were lysed using a high-throughput tissue grinder (Scientz, China) with glass beads in lysis ChIP buffer (50 mM HEPES-KOH [pH 7.5], 0.14 M NaCl, 1 mg/mL sodium deoxycholate, 1 mM EDTA, 1% Triton X-100, 1×protease inhibitor cocktail [Roche Diagnostics Gmbh, Germany], and 1 mM phenylmethylsulfonyl fluoride [PMSF]). Afterward, chromatin fragments of ~150 bp in size were acquired through sonicating in Diagenode Bioruptor Plus (Diagenode, Belgium). DNA was precipitated in the presence or absence of Set5 antibodies (Affinity Biosciences LTD, China) and incubated with 20 $\mu$L protein G beads (GE Healthcare, USA) at 4°C and 20 rpm overnight. Protein G beads were subsequently washed by the following buffers: (i) lysis ChIP buffer, (ii) wash buffer I (0.05 M HEPES-KOH [pH 7.5], 0.5 M NaCl, 1 mg/mL sodium deoxycholate, 1 mM EDTA, and 1% Triton X-100), (iii) wash buffer II (10 mM Tris [pH 8.0], 0.5% NP-40, 0.25 M LiCl, 5 mg/mL sodium deoxycholate, and 1 mM EDTA), and (iv) Tris-EDTA (TE) buffer. Protein and DNA were isolated from beads via a vortex oscillator at 1,000 rpm and 70°C for 30 min in 1% SDS-TE buffer. Finally, precipitated DNA, which interacted with Set5, was de-cross-linked and purified. ChIP enrichment was measured by real-time qPCR. The real-time quantitative primers used for target genes are listed in Table S3.

**Yeast two-hybrid interaction assay.** AH109 (Clontech, Japan) was used for host strain in yeast two-hybrid interaction assays. Parent plasmids pGADT7 and pGBKT7 (Clontech, Japan) acted as activation domain (AD) and DNA binding domain (BD), respectively. Target gene fragments, amplified from the genome of PLY01, were fused to pGADT7 or pGBKT7 through ClonExpress Ultra one-step cloning kit (Vazyme, China). The premiers for plasmids construction are listed in Table S3. Bait and prey plasmids were simultaneously introduced into AH109 using the lithium acetate method as mentioned and SD medium lacking tryptophan and leucine (SD-TL) for the selection of transformants. Yeast plates were incubated at 30°C for 3 to 5 days. The yeast strain AH-AMcm2-BMcm10 was employed as a positive control (PC), whereas AH-AHog1-B and AH-A-BSet5 were employed as negative control (NC).

**Western blot analysis.** Western blot analysis was performed as previously described with some modifications (47). Briefly, equal numbers of cells were collected when strains grew to the exponential phase in fermentation with acetic acid stress. Cells were treated with the following buffers: (i) distilled water, (ii) 2 M sodium hydroxide-8% mercaptoethanol, and (iii) TAP extraction buffer (4 mM HEPES-KOH [pH 7.5], 10% glycerol, 0.35 M NaCl, 0.1% Tween 20, 1× protease inhibitor cocktail, 1 mM PMSF, and 1× phosphatase inhibitor complex III [Sangon Biotech, China]). The sediment of cells was resuspended in 2× SDS loading buffer (0.1 M Tris-HCl [pH 6.8], 4% SDS, 0.2% bromophenol blue, 20% glycerol, and 2% mercaptoethanol) and boiled for 10 min. Protein samples were separated by 12% SDS-PAGE and transferred to Immobilon-P polyvinylidene difluoride (PVDF) membrane. The blots were probed with antibodies against specific proteins, followed by incubation with goat anti-mouse or goat anti-rabbit secondary antibodies. The antibodies used in this study are listed in Table S4. The specific proteins were visualized by super ECL detection reagent (Yeasen, China) in FluorChem FC3 (Proteinsimple, USA).

**Statistical analysis.** All experiments were independently performed in triplicate, and reproducible results were obtained. The data are shown as mean and standard deviation (SD). Statistical analysis was performed using Student's *t* test with a significant level of *P* value of <0.05.

**Data availability.** The phosphoproteomic data are also available from the PRIDE database under the accession number PXD028217.

## SUPPLEMENTAL MATERIAL

Supplemental material is available online only.
**SUPPLEMENTAL FILE 1**, PDF file, 0.9 MB.

## ACKNOWLEDGMENTS

This work was supported by the National Natural Science Foundation of China (no. 21978168), the National key research and development program (No. 2022YFE0108500), and Open Funding Project of the State Key Laboratory of Biocatalysis and Enzyme Engineering.

We appreciate the kind help of Jin-Qiu Zhou and Jia-Cheng Liu at Shanghai Institute of Biochemistry and Cell Biology, Chinese Academy of Sciences, for ChIP analysis.

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
