## [Reviewer comments · Microbiology Spectrum]

Microbiology Spectrum

Modification of phosphorylation sites in the yeast lysine methyltransferase Set5 exerts influences on the mitogen-activated protein kinase Hog1 under prolonged acetic acid stress

Xin-Qing Zhao, Pei-Liang Ye, Bing Yuan, Xue-Qing Wang, and Mingming Zhang

Corresponding Author(s): Xin-Qing Zhao, Shanghai Jiao Tong University

Review Timeline:

Submission Date:	August 14, 2022
Editorial Decision:	September 28, 2022
Revision Received:	October 31, 2022
Editorial Decision:	November 20, 2022
Revision Received:	January 19, 2023
Editorial Decision:	February 6, 2023
Revision Received:	February 13, 2023
Accepted:	February 27, 2023

Editor: Jing Han

Reviewer(s): Disclosure of reviewer identity is with reference to reviewer comments included in decision letter(s). The following individuals involved in review of your submission have agreed to reveal their identity: Angel Andrade Torres (Reviewer #1)

Transaction Report:

DOI: <https://doi.org/10.1128/spectrum.03011-22>

September 28, 2022

Prof. Xin-Qing Zhao
Shanghai Jiao Tong University
School of Life Science and Biotechnology
State Key Laboratory of Microbial Metabolism
Dongchuan Road 800
Shanghai, Shanghai 200240
China

Re: Spectrum03011-22 (Phosphorylation of the yeast lysine methyltransferase Set5 regulates the mitogen-activated protein kinase Hog1 under prolonged acetic acid stress)

Dear Prof. Xin-Qing Zhao:

Link Not Available

Sincerely,

Jing Han

Journals Department
Reviewer comments:

Reviewer #1 (Comments for the Author):

Ye et al. have investigated how amino acid substitutions within Set5 CTR domain alter yeast acetic acid stress response. Main finding is that Set5 affects Hog1 presumably by regulating it transcriptionally. My major concern is that authors claim that Phosphorylation of Set5 is associated with such regulation despite that actual phosphorylation of Set5 was not assessed at any point. Thus, authors cannot claim for such modification as responsible for none of the observed differences (Title plus L.103; 126-128; L145-46; L221-22; L273-274, L279, etc). Neither referred to Set5 with alanine substitution as "unphosphorylated".

There is also a lack of evidence that introduction of single or multiple (up to 10) substitutions did not alter Set5 stability/folding. It would be convenient (at minimum) to compare Set5 WT and multiple residue mutant's presence under tested conditions. An important part of the study is based on ChIP assays by using a custom-made (from this study) Set5 antibody. Therefore, I consider important to demonstrate specificity of this antibody since cross-reactivity might derive in wrong conclusions.

Other concerns,

L. 104. What phosphoproteomic data are referring?

L192-193. Presented data at this point is not sufficient to assert that "Set5 indeed regulates HOG1"

I could not find Supplementary Figures File. Due to most of supplementary figures are critical it makes the manuscript less readable.

Figure 2. Since at 20 min, most tested genes are downregulated (related to PLY01). Also, OLE1 is upregulated in all cases, no matter Set5 incorporated change (alanine or aspartic acid); substitution of serine residues seems to derive in a more effective activity of Set5 as repressor rather than to be specifically related with its phosphorylation.

Figure 2. Panels B-D and F-H are unnecessary. Also, data indicate that serine substitutions by aspartic acid or alanine does not have an impact in most evaluated genes (4/30).

Figure 3 B and D. There are not clear differences between Set5 WT vs multiple residue mutants (either with alanine or aspartate substitutions). For instance, all Set5 mutants shown a reduction in its interaction with HOG1. And, the opposite result was obtained under no stress growth conditions. This contradictory result is not addressed. Also it points out to an additional Set5 partner, promoting Set5 binding to HOG1 during acetic acid stress. It is relevant to compare the protein levels of Set5 in both evaluated conditions since observed differences in PLY01 might derive of higher Set5 amounts during growth in acetic acid stress.

Why ChIP-qPCR evaluations were achieved at different acetic acid concentrations (Fig. 3B, 7.5 g/l vs 3C, 5.0 g/L)?

Figure 4. Phosphorylated Hog-1 levels are conspicuously different when comparing PLY01 in panels A vs C.

Also, Why Hog1 phosphorylation was assessed at different acetic acid concentrations (Fig. 4A, 5.0 g/l vs 4C, 7.5 g/L)?

L. 301-302. There is no data showing interaction between Set5 and Ak11 or Rim15.

Authors must discuss why lower ROS accumulations was observed after Set5 aspartate substitutions (compared to alanine changes).

There are minor typos throughout

Reviewer #2 (Comments for the Author):

The authors investigated the role of Set5 in response to acetic acid stress tolerance. They found that phosphorylation of Set5 during acetic acid stress modulated the expression of Hog1. The study was interesting and hypothesis-generating in nature.

Minor Comments:

1. It was unclear as to how the transformants generated in this study were correctly verified.
2. Mass-spectrometry-based phosphoproteomics would be a useful to validate the findings of this paper. It would have been nice to see this technique included.
3. Overall, the data supports the findings of the study.
4. There are few spelling errors throughout the paper. Please correct these before publication.

Staff Comments:

Preparing Revision Guidelines

Please return the manuscript within 60 days; if you cannot complete the modification within this time period, please contact me. If you do not wish to modify the manuscript and prefer to submit it to another journal, please notify me of your decision immediately so that the manuscript may be formally withdrawn from consideration by Microbiology Spectrum.

Point-by-point responses to the reviewers' comments

Response to Reviewer #1:

Ye et al. have investigated how amino acid substitutions within Set5 CTR domain alter yeast acetic acid stress response. Main finding is that Set5 affects Hog1 presumably by regulating it transcriptionally. My major concern is that authors claim that Phosphorylation of Set5 is associated with such regulation despite that actual phosphorylation of Set5 was not assessed at any point. Thus, authors cannot claim for such modification as responsible for none of the observed differences (Title plus L.103; 126-128; L145-46; L221-22; L273-274, L279, etc). Neither referred to Set5 with alanine substitution as "unphosphorylated".

Authors' response: We sincerely thank the reviewer for the helpful comments and suggestions, which help to significantly improve our manuscript. We agree with the reviewer and have revised all the expression of 'phosphorylation' in the title as well as the main text to 'modification of the phosphorylation sites'. We also revised the text accordingly in the manuscript. In addition, we have changed all the description of "unphosphorylated" to "alanine substitution" in the text.

In our unpublished studies, we have tried heterologous expression of Set5 in *E. coli*, but only a very low expression level was obtained. As indicated in the main text, the preparation of antibody for detecting Set5 phosphorylation has been challenging and was not successful. We also did not find a related report of antibody in the literature. Therefore, we only performed *in vivo* studies in this work.

Aspartic acid substitution provides a negative charge that is similar to phosphate group, and has been commonly used to mimic the constitutive phosphorylation of proteins (EMBO J. 2022, 41(4): e108290; J Biol Chem. 2021, 296: 100164). On the contrary, alanine substitution could prevent the site from being phosphorylated and minimize the impact on the protein structure, which is generally used for searching key residues, the mutation of which leads to loss of kinase activity (mBio. 2022, 13(3): e0103422; FEBS Lett. 2021, 595(14):1886-1901; Mol Cell Biol. 2017, 37(16): e00655-16). Therefore, we assume that the genetic experiments in this study might support the effect of Set5 phosphorylation. We will further the studies on Set5 phosphorylation after obtaining suitable antibodies, and have modified the text in the revised manuscript.

There is also a lack of evidence that introduction of single or multiple (up to 10) substitutions did not alter Set5 stability/folding. It would be convenient (at minimum) to compare Set5 WT and multiple residue mutant's presence under tested conditions. An important part of the study is based on ChIP assays by using a custom-made (from this study) Set5 antibody. Therefore, I consider important to demonstrate specificity of this antibody since cross-reactivity might derivate in wrong conclusions.

Authors' response: We thank the reviewer for the helpful comments. We adopted mutation sites of multiple phosphorylation of Set5 based on the previous studies, where the Set5 derivative proteins carrying the mutations analyzed in our study were heterologously expressed in *E. coli* for *in vitro* methylation assays (Mol Cell Biol. 2020, 40(2): e00341-19). From the reported results in the studies, no stability issue was revealed. On the other hand, as stated above, the substitutions we used in this study are commonly used to study protein phosphorylation (EMBO J. 2022, 41(4): e108290; J Biol Chem. 2021, 296: 100164). However, we cannot rule out the possibility that the mutations may affect protein folding and/or stability of Set5. We have added discussion of this point in the revised manuscript (P. 16-17, Line 309-311).

We sincerely thank the reviewer for the helpful comments on the result. We confirmed that the Set5 mutated genes were correct using diagnostic PCR (Fig. 1), and different phenotypes we observed using the different mutants suggest the presence of the mutated proteins. We used custom-made Set5 for the investigation (Fig. R1), and we agree with the reviewer that the specificity of the antibody is important. We also confirmed the results of the ChIP assays using the anti-His antibody in another strain *S. cerevisiae* BY4741 (Fig S6), which supports the correctness of the ChIP assay results.

Fig. R1 Detection of Set5 using the Set5 antibody (indicated by the blue arrow)

Other concerns:

L. 104. What phosphoproteomic data are referring?

Authors' response: We are sorry for not introducing the source of phosphoproteomic data, which has been reported in our previous study published in Bioresource Technology (2022, 348:126758). The related information has been added in the revised manuscript. The phosphoproteomic data are also available from the PRIDE database under the accession number of PXD028217, and we have added this information in the manuscript (P.25, Line 487-488).

L192-193. Presented data at this point is not sufficient to assert that "Set5 indeed regulates HOG1"

Authors' response: We have deleted the sentence "Set5 indeed regulates *HOG1*" in the manuscript and added discussions, and we modified the title to state that Set5 exerts control on Hog1. We made the conclusion based on the below results: 1) Figure 2H showed that the transcriptional level of *HOG1* was enhanced in PLY01-SET5-10D when compared to PLY01-SET5-10A, which suggests that modification of the phosphorylation sites of Set5 leads to transcription changes of *HOG1*. 2) The results in Fig 3 and Fig 4 further confirmed that Set5 is enriched to *HOG1*, and modification of phosphorylation sites of Set5 affects the protein expression of Hog1. These results demonstrate that Set5 exerts control on Hog1. However, more studies are needed to reveal whether Set5 directly regulates *HOG1*. We have added related discussion in the revised manuscript (P. 15, Line 273-274).

I could not find Supplementary Figures File. Due to most of supplementary figures are critical it makes the manuscript less readable.

Authors' response: We are also confused why the supplemental figures were not present, we have submitted the file previously. We have confirmed that the Supplementary figures are present in the revised version of the manuscript.

Figure 2. Since at 20 min, most tested genes are downregulated (related to PLY01). Also, *OLE1* is upregulated in all cases, no matter Set5 incorporated change (alanine or aspartic acid); substitution of serine residues seems to derivate in a more effective activity of Set5 as repressor rather than to be specifically related with its phosphorylation.

Authors' response: We thank the reviewer for pointing out this issue. We agree with the reviewer that Set5 may serve as a repressor directly or indirectly for some genes, which may not be totally dependent on its phosphorylation states. We only tested limited genes in this study, and based on the results, we indeed found regulation of gene transcription by modification of Set5 phosphorylation sites. More studies will be performed to examine in detail the global effects of the modification.

Regulation of gene expression is normally affected by a complicated network. We hypothesize that mutations of the Set5 phosphorylation sites (alanine or aspartic acid) may cause fine-tuned regulation of downstream genes. For regulation of gene expression, neither too much nor too less phosphorylation of Set5 may not be beneficial, so the results caused by two different modification methods of the Set5 phosphorylation sites may not be opposite completely. We have discussed this point in the revised manuscript (P. 15, Line 280-292).

Figure 2. Panels B-D and F-H are unnecessary. Also, data indicate that serine

substitutions by aspartic acid or alanine does not have an impact in most evaluated genes (4/30).

Authors' response: We thank the reviewer for the helpful discussion. Fig. 2 B-D and F-H showed the comparisons between the phosphomimetic and alanine substitution Set5 mutant strains. The evaluated genes are chosen from the literature and also from our multi-omics analyses results (including transcriptome, proteome, and phosphoproteome) between SPSC01 and PLY01 under acetic acid stress (Bioresource Technology, 2022, 348:126758). We detected different Set5 phosphorylation levels in SPSC01 and PLY01 by phosphoproteomic analysis, and we thus assumed that these genes might be potentially regulated by Set5. We agree with the reviewer that in the current results, only some genes were affected by Set5 phosphorylation. However, if more genes will be investigated and more time points evaluated, we might find more changes in global gene transcription. The current results help us to better understand the effects of Set5 phosphosites modification, and promote further investigation on the effect of Set5 phosphorylation. Therefore, we would like to keep these figures. We updated the discussion in the revised manuscript (P. 16, Line 292-296).

Figure 3 B and D. There are not clear differences between Set5 WT vs multiple residue mutants (either with alanine or aspartate substitutions). For instance, all Set5 mutants shown a reduction in its interaction with *HOG1*. And, the opposite result was obtained under no stress growth conditions. This contradictory result is not addressed. Also it points out to an additional Set5 partner, promoting Set5 binding to *HOG1* during acetic acid stress. It is relevant to compare the protein levels of Set5 in both evaluated conditions since observed differences in PLY01 might derivate of higher Set5 amounts during growth in acetic acid stress.

Authors' response: In Fig. 3B and D, we observed clear differences between PLY01 and the mutants: under acetic acid stress, the mutants showed lower enrichment comparing with PLY01 (Fig. 3B), whereas under stress free conditions, enrichment is higher in the mutants (Fig. 3B). We hope that we make it clear for these results. So far, we cannot explain why there exist differences between PLY01 and the mutants, which deserve further examination.

We also feel that the opposite results are interesting. We assume that the whole regulatory network is different under acetic acid stress condition and the stress-free condition. For example, according to our unpublished data, under stress free conditions, overexpression of Set5 only led to limited changes in global protein expression (less than 10 proteins were differentially expressed), whereas under acetic acid stress condition, more than 300 proteins were remarkably changed by Set5 overexpression. It will be meaningful to study why Set5 works differently under the stress condition and the stress-free condition. We appreciate the wonderful comments

from the reviewer, which give us a new inspiration that there might be an additional Set5 partner to promote Set5 binding to *HOG1* during acetic acid stress. We will further test this hypothesis in our future work. We have added discussion in the revised manuscript (P. 16, Line 297-307).

We performed the ChIP assay by the supervision of Prof. Jinqiu Zhou following the method in their work (Nucleic Acids Res. 2020, 48(22):12792-12803), and we believe that the results are reliable. In the ChIP assay, the two Set5 mutants and PLY01 are all sampled under the same condition of acetic acid stress, and we assumed that the replacement of serine with alanine or aspartate had no impact on the amount of Set5 protein level, because this is a classical method for phosphorylation-related studies as discussed above (EMBO J. 2022, 41(4): e108290; J Biol Chem. 2021, 296: 100164). Due to the challenge of antibody, we did not check the protein level here. However, we agree with the reviewer that more delicate studies are needed to investigate the protein levels. We have added related discussion in the manuscript (P. 16, Line 307-311).

Why ChIP-qPCR evaluations were achieved at different acetic acid concentrations (Fig. 3B, 7.5 g/l vs 3C, 5.0 g/L)?

Authors' response: SPSC01 showed better growth than PLY01 under 5.0 g/L acetic acid, therefore, for all the comparisons between SPSC01 and PLY01, we used 5.0 g/L acetic acid. However, there is no significant growth difference between PLY01, PLY01-SET5-S458A, and PLY01-SET5-S458D at 5.0 g/L acetic acid. Therefore, the tested concentration of acetic acid was increased to 7.5 g/L in the subsequent experiments to better distinguish the differences between PLY01 and Set5 mutation strains. We have included this information in the revised text (P. 11, Line 189-192).

Figure 4. Phosphorylated Hog-1 levels are conspicuously different when comparing PLY01 in panels A vs C.

Authors' response: Yes, as discussed above, when comparing SPSC01 and PLY01, 5 g/L acetic acid was used, whereas when comparing the different PLY01 mutants, we used 7.5 g/L acetic acid. The higher level of phosphorylated Hog1 of PLY01 in the presence of 7.5 g/L acetic acid compared to 5.0 g/L acetic acid suggests that more severe acetic acid stress drives higher levels of Hog1 phosphorylation (A vs C). We have added this information in the results (P. 13, Line 233-236).

Also, Why Hog1 phosphorylation was assessed at different acetic acid concentrations (Fig. 4A, 5.0 g/l vs 4C, 7.5 g/L)?

Authors' response: As described above, SPSC01 showed better growth than PLY01 under 5.0 g/L acetic acid. However, there is no significant growth difference between PLY01, PLY01-SET5-S458A, and PLY01-SET5-S458D under this concentration of

acetic acid (Fig. S1). Therefore, the tested concentration of acetic acid was increased to 7.5 g/L in the subsequent experiments to better distinguish the differences between PLY01 and Set5 mutation strains.

L. 301-302. There is no data showing interaction between Set5 and Ak11 or Rim15.

Authors' response: We are sorry for the mistake in uploading the supplemental figures, which have been added in the revised submission. The results were shown in Fig. S7, which are also presented below (Fig. R1).

Fig. R1 Interaction of Set5 with various kinases

Authors must discuss why lower ROS accumulations was observed after Set5 aspartate substitutions (compared to alanine changes).

Authors' response: Yes, we propose a possible mechanism underlying lower ROS accumulation by the Set5 aspartate substitutions: 1) aspartate substitutions mimic phosphorylation of Set5; 2) phosphorylated Set5 enriches to *HOG1* sequence and improves *HOG1* transcription; 4) More *HOG1* mRNA leads to enhanced abundance of Hog1/Hog-P; 5) the activated Hog1 regulates transcription of the catalase gene *CTT1* or affects the expression of Gpx1; 6) Improved antioxidant enzyme activities result in more active scavenging ROS induced by acetic acid, leading to lower ROS accumulation. We have added the above hypothesis in the revised manuscript (P. 17, Line 326-333).

There are minor typos throughout.

Authors' response: We thank the reviewer for the comment. We have carefully revised the English writing of the manuscript.

Response to Reviewer #2:

The authors investigated the role of Set5 in response to acetic acid stress tolerance. They found that phosphorylation of Set5 during acetic acid stress modulated the expression of Hog1. The study was interesting and hypothesis-generating in nature.

Authors' response: We thank the reviewer for the positive comments.

Minor Comments:

1. It was unclear as to how the transformants generated in this study were correctly verified.

Authors' response: We have verified each strain by sequencing the Set5 gene, and the primers V-SET5-F (GGTACAACACTGTACATACTGAAGAG) and V-SET5-R (CGCTCTCAAATTCACCTCTCA) are used for PCR amplification. We have added this information in the revised manuscript (P. 20-21, Line 394-397).

2. Mass-spectrometry-based phosphoproteomics would be a useful to validate the findings of this paper. It would have been nice to see this technique included.

Authors' response: We agree with the reviewer that the phosphoproteome studies contribute to our findings in this work. We have analyzed the phosphoproteomic data in our previous paper (Bioresource Technology 2022, 348:126758). We have added this information in the revised manuscript (P. 7, Line 104-106).

3. Overall, the data supports the findings of the study.

Authors' response: We thank the reviewer for the positive comments.

4. There are few spelling errors throughout the paper. Please correct these before publication.

Authors' response: We thank the reviewer for the comment. We have carefully revised the English writing of the manuscript.

November 20, 2022

Prof. Xin-Qing Zhao
Shanghai Jiao Tong University
School of Life Science and Biotechnology
State Key Laboratory of Microbial Metabolism
Dongchuan Road 800
Shanghai, Shanghai 200240
China

Re: Spectrum03011-22R1 (Modification of phosphorylation sites in the yeast lysine methyltransferase Set5 exerts influences on the mitogen-activated protein kinase Hog1 under prolonged acetic acid stress)

Dear Prof. Xin-Qing Zhao:

Link Not Available

Sincerely,

Jing Han

Journals Department
Reviewer comments:

Reviewer #1 (Comments for the Author):

I thank the authors for revising their manuscript. It reads better. In regard to my concern about Set5 antibody specificity (employed in ChIP assays), cross reactive bands are observed in the Figure: "Fig. R1 Detection of Set5 using the Set5 antibody", attached to their response letter. Accordingly, supporting data by using the anti-His antibody in strain BY4741 is relevant.

Reviewer #3 (Comments for the Author):

The authors provide new details about how *Saccharomyces cerevisiae* responds to excess acetic acid stress. They show that mutation of potential phosphorylation sites in methyl transferase Set5 changes acetic acid tolerance. They used these same Set5 mutants to discover that some RNA transcripts from a set of 31 genes tested increased or decreased. They additionally demonstrated that Set5 accumulated on HOG1 DNA and that Set5 and Hog1 proteins interact via the two-hybrid assay. Together these results show how Set5 fits into the Hog1 protein kinase network to reduce acetic acid induced reactive oxygen species.

While the findings in the paper are informative, the writing suffered from much imprecision and poor sentence construction. I list many suggestions for improvement below (there are many more to find). Scientifically, the issues I raise on Lines 149-154 and Figure S6 are the most important to address.

Line 20, 192 and 281: The phrase "in-frame region" is cryptic. Do you mean "coding region"?

Line 25: Change "implied" to imply"

Line 29: "modification of Set5 phosphorylation sites" is ambiguous, be specific. What changes have you tested? Further on in that same sentence, "exerts control" could be replaced with something more specific.

Line 40: The way this sentence is written, it is impossible to tell if acetic acid or lignocellulosic biomass is the major toxic substance. Because it is acetic acid, put the "which" clause right after acetic acid.

Line 91-93: Change to "Furthermore, proteins homologous to Set5 are associated with bone morphogenesis, cancer, and various other diseases in humans and other eukaryotes. "

Line 97 repeats part of line 86.

Line 101: Change "revealed" to "reveal". Line 110: Change "folds" to "fold". Line 113: Omit "*S. cerevisiae*"

Line 115: The sentence "A total of 6 mutants corresponding to the three phosphosites S458, S461, and S462 were constructed by CRISPR/Cas9-mediated genome editing." Is out of place because the next sentence refers to Fig. S1, which only has the S2458A and S458D mutants.

Lines 149-154 discuss data in Fig S4A, which shows expression levels of 31 selected genes in the SET5 mutants. The presentation of the data in Fig S4A is difficult to parse because the graphics bars are so close together. If there was more space between the bars for each gene, it would improve reading that graph. Also, a line across at $y=1.0$ would make it easier to see which genes had increased or reduced expression. Furthermore, the "obviously affected" on line 153 should include some statistic for comparison to the changes in other genes to know statistical significance because there were many other genes changing besides the ones discussed. Moreover, throughout this discussion of that data, they stated transcript level changes were "affected", it would be better to be specific by saying "increased" or "decreased".

Line 169 and 445: Change "20/60" to "20 or 60 min".

Line 171: Insert "whether" into "no matter alanine or aspartic" to make "no matter whether alanine or aspartic".

Line 184: Omit "in the phosphoproteomic data" because it is redundant.

Figure S6 figure legend needs more detail. What is the Y-axis? The HOG blue bar needs coordinates and "AUG" added so that is clear where it starts. It seems strange that only the interval between yellow dotted lines has a significant difference when there are many regions to the left, 5' to that interval that the mutant had binding and the wild-type none. Additionally, why are the two peaks in wild-type near the initiation codon and promoter not significant and considered in the regulation of HOG1?

Figure 3: The meaning of "A.U." for enrichment in Figure 3 is not defined. Do you mean fold enrichment of mutant versus wild-type?

Many times, in the discussion, there was unnecessary mention that these are the first discovery of various findings.

The figure 6 summary slide does not include the physical interaction of Set5 and Hog1 into the scheme. The figure legend also states that Set5 "specifically" promotes HOG1 transcription whereas figures 2 and S4 show that RNA levels of many genes were affected by Set5 phosphorylation.

Line 402: Change "engineering" to "engineered".

Line 438-439: Say "The column was eluted with 4 mM sulfuric acid, at a 0.6 mL/min flow rate, at 50 {degree sign}C."

Lines 499-500: Change "triplicates" to "triplicate". What are the criteria of "reproducible results"?

Reviewer #4 (Comments for the Author):

The manuscript "Modification of phosphorylation sites in the yeast lysine methyltransferase Set5 exerts influences on the mitogen-activated protein kinase Hog1 under prolonged acetic acid stress" by Ye et.al. describes a series of experiments to investigate the mechanism by which Set5 promotes tolerance to acetic acid stress. The authors indicate that Set5 phosphorylation is elevated in response to acetic acid stress and this is correlated with alterations in global gene expression. Mutation of a series of serine and threonine residues in the endogenous Set5 carboxyl-terminus to alanine or aspartic acid was performed. While the response of the cells, as measured by growth and fermentation capability, was variable, evidence is presented to suggest that these phosphorylation sites influence the ability of Set5 to support tolerance to acetic acid stress. The mitogen activated protein kinase Hog1 is among the genes induced by acetic acid stress and the authors provide Chromatin Immunoprecipitation data to show that Set5 binds to the HOG1 gene with enrichment across the coding sequence. The authors further provide yeast two-hybrid data supporting the contention that Hog1 and Set5 physically interact.

1. The residues phosphorylated have been reported by others as well. There is no data presented to support the statement that phosphorylation is increased by acetic acid stress.
2. The conclusion that phosphorylation of the Set5 carboxyl-terminus seems to be based on the idea that the Aspartic acid (presumably phosphomimetic) mutants perform better than the alanine mutants that are non-phosphorylatable. Based on figure 1 the 10A and 10D mutants are not different and the S458D mutant alone performs better than either of the multiple mutants and 3A performs as well as 10D. While it is reasonable to conclude the mutations are having an effect it is difficult to draw a strong conclusion that it is the phosphorylation of these residues that is responsible for the phenotypes and gene expression effects observed.
3. Validation of the Set5 antibody would be extremely valuable as well as a simple western blot to determine if the expression of Set5 is altered by the mutations. These changes could for example stabilize the protein which would be important information and could help shape the investigators understanding of the mechanism by which Set5 influences acetic acid tolerance.
4. No methods are provided for the chromatin IP experiments that localize Set5 to the coding sequence of HOG1. I assume this is some form of ChiP-seq but methods and control data need to be included.
5. Figure 1 and Figure S2 appear to be the same data, just compressed in S2 so this has no value.
6. Figure 4B and 4D should be labelled with relative abundance rather than relatively changed abundance. The western blots in Figure 4A and 4C are used to make quantitative arguments. The use of ECL is not favorable for quantitative analysis but is likely acceptable in this context. It is notable that in the reference provided for the western blots (Mei, et.al.) fluorescently tagged secondary antibodies were used as is now convention for quantitative blotting. Minimally the authors should indicate whether duplicate blots were probed for Hog1, phospho-Hog1 as may be the case here since not all of the Hog1 and Hog1-p bands seem to align well, or if a single blot was probed, stripped, and then re-probed.
7. The discussion lines 313 - 315 state that results imply that there may be an additional Set5 partner. It is not clear to me what results are being referred to here and the authors should be more explicit.
8. Line 48 - "...prevention of harmful fungi." it is not clear what this means.
9. Line 54, 55 The authors state that "...flocculation regulates tolerance to ethanol, furfural and acetic acid. " Since the mechanism is not understood it would be more correct to indicate that these things are correlated or associated rather than one regulates the other.
10. Similarly line 216 attributes enrichment of Set5 binding to HOG1 to flocculation. It would be more correct to indicate that the Set5 binding to the HOG1 coding sequence is enriched in the flocculating strain. This is not trivial since flocculation is more likely a consequence of a genetic change that is associated with the enrichment rather than the cause of it.
11. Line 163: "... whose expression improved multiple stresses..." should be "... whose expression improved tolerance to multiple stresses..."

Staff Comments:

Preparing Revision Guidelines

Please return the manuscript within 60 days; if you cannot complete the modification within this time period, please contact me. If you do not wish to modify the manuscript and prefer to submit it to another journal, please notify me of your decision

immediately so that the manuscript may be formally withdrawn from consideration by Microbiology Spectrum.

Point-to-Point Responses to the Reviewers' comments

Reviewer #1

I thank the authors for revising their manuscript. It reads better. In regard to my concern about Set5 antibody specificity (employed in ChIP assays), cross reactive bands are observed in the Figure: "Fig. R1 Detection of Set5 using the Set5 antibody", attached to their response letter. Accordingly, supporting data by using the anti-His antibody in strain BY4741 is relevant.

Authors' response: We thank the reviewer for the helpful discussions and comments.

Reviewer #3

The authors provide new details about how *Saccharomyces cerevisiae* responds to excess acetic acid stress. They show that mutation of potential phosphorylation sites in methyl transferase Set5 changes acetic acid tolerance. They used these same Set5 mutants to discover that some RNA transcripts from a set of 31 genes tested increased or decreased. They additionally demonstrated that Set5 accumulated on HOG1 DNA and that Set5 and Hog1 proteins interact via the two-hybrid assay. Together these results show how Set5 fits into the Hog1 protein kinase network to reduce acetic acid induced reactive oxygen species.

While the findings in the paper are informative, the writing suffered from much imprecision and poor sentence construction. I list many suggestions for improvement below (there are many more to find). Scientifically, the issues I raise on Lines 149-154 and Figure S6 are the most important to address.

Line 20, 192 and 281: The phrase "in-frame region" is cryptic. Do you mean "coding region"?

Authors' response: Yes, it is coding region, we have revised the phrase in the main text.

Line 25: Change "implied" to imply"

Authors' response: Yes, revised.

Line 29: "modification of Set5 phosphorylation sites" is ambiguous, be specific. What changes have you tested? Further on in that same sentence, "exerts control" could be replaced with something more specific.

Authors' response: We modified the phosphorylation sites using the substitution of

Aspartic acid to mimic the constitutive phosphorylation of proteins (EMBO J. 2022, 41: e108290; J Biol Chem. 2021, 296: 100164), and alanine substitution to mimic loss of kinase activity (mBio. 2022, 13: e0103422; FEBS Lett. 2021, 595(14):1886-1901; Mol Cell Biol. 2017, 37: e00655-16) as classically used to study protein phosphorylation. Based on the comment of the other reviewer, we modified the sentence as: the mutants carrying the modified Set5 phosphorylation sites showed altered expression and phosphorylation of Hog1 in the revised manuscript.

Line 40: The way this sentence is written, it is impossible to tell if acetic acid or lignocellulosic biomass is the major toxic substance. Because it is acetic acid, put the "which" clause right after acetic acid.

Authors' response: We agree with the reviewer and revised the sentence.

Line 91-93: Change to Furthermore, proteins homologous to Set5 are associated with bone morphogenesis, cancer, and various other diseases in humans and other eukaryotes.

Authors' response: We thank the reviewer and revised the sentence as suggested (Line 88-90 in the revised manuscript).

Line 97 repeats part of line 86.

Authors' response: Yes, we have revised the sentence (Line 97-99).

Line 101: Change "revealed" to "reveal". Line 110: Change "folds" to "fold". Line 113: Omit "S. cerevisiae"

Authors' response: Yes, revised.

Line 115: The sentence "A total of 6 mutants corresponding to the three phosphosites S458, S461, and S462 were constructed by CRISPR/Cas9-mediated genome editing." Is out of place because the next sentence refers to Fig. S1, which only has the S2458A and S458D mutants.

Authors' response: Yes, revised.

Lines 149-154 discuss data in Fig S4A, which shows expression levels of 31 selected genes in the SET5 mutants. The presentation of the data in Fig S4A is difficult to parse because the graphics bars are so close together. If there was more space between the bars for each gene, it would improve reading that graph. Also, a line across at y=1.0 would make it easier to see which genes had increased or reduced expression. Furthermore, the "obviously affected" on line 153 should include some statistic for

comparison to the changes in other genes to know statistical significance because there were many other genes changing besides the ones discussed. Moreover, throughout this discussion of that data, they stated transcript level changes were "affected", it would be better to be specific by saying "increased" or "decreased".

Authors' response: We thank the reviewer for the helpful suggestion. We have added a line at $y=1.0$ to show the critical values. Most of the genes did not show very significant changes in Figure 4A (less than 1.5-fold), and we have added the details of the significantly changed genes in the main text (Line 151-152).

Line 169 and 445: Change "20/60" to "20 or 60 min".

Authors' response: Yes, revised.

Line 171: Insert "whether" into "no matter alanine or aspartic" to make "no matter whether alanine or aspartic".

Authors' response: Yes, revised.

Line 184: Omit "in the phosphoproteomic data" because it is redundant.

Authors' response: Yes, revised.

Figure S6 figure legend needs more detail. What is the Y-axis? The HOG blue bar needs coordinates and "AUG" added so that is clear where it starts. It seems strange that only the interval between yellow dotted lines has a significant difference when there are many regions to the left, 5' to that interval that the mutant had binding and the wild-type none. Additionally, why are the two peaks in wild-type near the initiation codon and promoter not significant and considered in the regulation of HOG1?

Authors' response: We regret for the previous unclear description. We have modified Figure S6 and the legend. The x-axis shows the position in the chromosome, and the y-axis of the peak signal indicates the extent of enrichment. We added AUG in the revised figure. In the previous study, we used our unpublished data comparing the difference between Set5 and its mutant with deletion of the zinc finger region. Considering that the work in this manuscript is not related to the effect of zinc finger, we deleted the data of the mutant, and we will report the effect of the zinc finger region on Set5 enrichment of the target genes in another separate article. We obtained data showing that only the region between the yellow lines is significant in the enrichment of Set5, and we used this region for the current study.

Figure 3: The meaning of "A.U." for enrichment in Figure 3 is not defined. Do you mean fold enrichment of mutant versus wild-type?

Authors' response: A.U. is shorted for arbitrary units, which is the strength of the enrichment signal for each sample. We have defined it in the legend.

Many times, in the discussion, there was unnecessary mention that these are the first discovery of various findings.

Authors' response: We have revised the discussion to avoid such writings.

The figure 6 summary slide does not include the physical interaction of Set5 and Hog1 into the scheme. The figure legend also states that Set5 "specifically" promotes HOG1 transcription whereas figures 2 and S4 show that RNA levels of many genes were affected by Set5 phosphorylation.

Authors' response: We have added physical interaction in the revised Fig. 6. We thank the reviewer and delete "specifically" in the legend (Line 729-730).

Line 402: Change "engineering" to "engineered".

Authors' response: Yes, revised.

Line 438-439: Say "The column was eluted with 4 mM sulfuric acid, at a 0.6 mL/min flow rate, at 50 {degree sign}C."

Authors' response: Yes, revised.

Lines 499-500: Change "triplicates" to "triplicate". What are the criteria of "reproducible results"?

Authors' response: Yes, revised.

Reviewer #4

The manuscript "Modification of phosphorylation sites in the yeast lysine methyltransferase Set5 exerts influences on the mitogen-activated protein kinase Hog1 under prolonged acetic acid stress" by Ye et.al. describes a series of experiments to investigate the mechanism by which Set5 promotes tolerance to acetic acid stress. The authors indicate that Set5 phosphorylation is elevated in response to acetic acid stress and this is correlated with alterations in global gene expression. Mutation of a series of serine and threonine residues in the endogenous Set5 carboxyl-terminus to alanine or aspartic acid was performed. While the response of the cells, as measured by growth and fermentation capability, was variable, evidence is presented to suggest

that these phosphorylation sites influence the ability of Set5 to support tolerance to acetic acid stress. The mitogen activated protein kinase Hog1 is among the genes induced by acetic acid stress and the authors provide Chromatin Immunoprecipitation data to show that Set5 binds to the HOG1 gene with enrichment across the coding sequence. The authors further provide yeast two-hybrid data supporting the contention that Hog1 and Set5 physically interact.

1. The residues phosphorylated have been reported by others as well. There is no data presented to support the statement that phosphorylation is increased by acetic acid stress.

Authors' response: We have reported the phosphoproteomic data in our previous study published in *Bioresource Technology* (2022, 348:126758). The related information has indicated in the manuscript (Line 106-108).

2. The conclusion that phosphorylation of the Set5 carboxyl-terminus seems to be based on the idea that the Aspartic acid (presumably phosphomimetic) mutants perform better than the alanine mutants that are non-phosphorylatable. Based on figure 1 the 10A and 10D mutants are not different and the S458D mutant alone performs better than either of the multiple mutants and 3A performs as well as 10D. While it is reasonable to conclude the mutations are having an effect it is difficult to draw a strong conclusion that it is the phosphorylation of these residues that is responsible for the phenotypes and gene expression effects observed.

Authors' response: We thank the reviewer for the helpful comment. Due to the complexity of signal transduction and regulation, we cannot explain thoroughly the mechanism in the current work, but we indeed observed changes led by modification of the phosphorylation sites. We agree with the reviewer that it is still not clear whether it is the direct effect or indirect effect in the changes, and have added this notion in the abstract (Line 30-31) and discussion (Line 281-282, Line 345). We also modified the title as: Phosphorylation of the yeast lysine methyltransferase Set5 is associated with the differential expression and phosphorylation of the mitogen-activated protein kinase Hog1 under prolonged acetic acid stress.

3. Validation of the Set5 antibody would be extremely valuable as well as a simple western blot to determine if the expression of Set5 is altered by the mutations. These changes could for example stabilize the protein which would be important

information and could help shape the investigators understanding of the mechanism by which Set5 influences acetic acid tolerance.

Authors' response: We appreciate very much the suggestion from the reviewer. However, we regret that the Set5 antibody by custom preparation worked not very well in our experiment. We will further pursuit whether Set5 mutations in this study affect protein level of Set5. However, we assumed that the replacement of serine with alanine or aspartate had no impact on the Set5 protein level, because this is a classical method for phosphorylation-related studies (EMBO J. 2022, 41: e108290; J Biol Chem. 2021, 296: 100164), and in the literature there is no report that the modifications affected protein stability. We believe that our conclusion is not affected without detection of Set5. It will be important to further study the in-depth mechanism, and we agree with the reviewer that it will be nice to perform the study in the future. We have discussed this notion in Line 310-313.

4. No methods are provided for the chromatin IP experiments that localize Set5 to the coding sequence of HOG1. I assume this is some form of ChIP-seq but methods and control data need to be included.

Authors' response: Yes, it is from the ChIP-seq data. We have added details in the supplementary figure legend of Fig.S6.

5. Figure 1 and Figure S2 appear to be the same data, just compressed in S2 so this has no value.

Authors' response: We evaluated short-term effect in Figure 1 and long-term effect in Figure S2. We have revised in the legend to make it clear.

6. Figure 4B and 4D should be labelled with relative abundance rather than relatively changed abundance. The western blots in Figure 4A and 4C are used to make quantitative arguments. The use of ECL is not favorable for quantitative analysis but is likely acceptable in this context. It is notable that in the reference provided for the western blots (Mei, et.al.) fluorescently tagged secondary antibodies were used as is now convention for quantitative blotting. Minimally the authors should indicate whether duplicate blots were probed for Hog1, phospho-Hog1 as may be the case here since not all of the Hog1 and Hog1-p bands seem to align well, or if a single blot was probed, stripped, and then re-probed.

Authors' response: We appreciate very much the comment from the reviewer. We

would like to keep the way for that the comparison in Figure 4B and 4D. We performed the experiments three times, and the results are the same.

7. The discussion lines 313 - 315 state that results imply that there may be an additional Set5 partner. It is not clear to me what results are being referred to here and the authors should be more explicit.

Authors' response: As shown in the ChIP-seq data, the peaks indicating Set5 enrichment in the coding region of *HOG1* is not so narrow. Considering that Set5 is not a transcription factor, but instead a methyltransferase, we assume that there may exist an additional Set5 partner which promotes Set5 binding to the target genes of Set5, that is, Set5 may work together with other protein(s) for regulating *HOG1* transcription, which will be the target of our future studies.

8. Line 48 - "...prevention of harmful fungi." it is not clear what this means.

Authors' response: We would like to state the fungi which make the food spoiled. We have revised this sentence.

9. Line 54, 55 The authors state that "...flocculation regulates tolerance to ethanol, furfural and acetic acid. " Since the mechanism is not understood it would be more correct to indicate that these things are correlated or associated rather than one regulates the other.

Authors' response: Yes, we have revised here and changed to "associated with enhanced tolerance..." (Line 55).

10. Similarly line 216 attributes enrichment of Set5 binding to *HOG1* to flocculation. It would be more correct to indicate that the Set5 binding to the *HOG1* coding sequence is enriched in the flocculating strain. This is not trivial since flocculation is more likely a consequence of a genetic change that is associated with the enrichment rather than the cause of it.

Authors' response: We agree with the reviewer and have changed the description (Line 214-215).

11. Line 163: "... whose expression improved multiple stresses..." should be "... whose expression improved tolerance to multiple stresses..."

Authors' response: Yes, we have revised the sentence as suggested by the reviewer.

February 6, 2023

Prof. Xin-Qing Zhao
Shanghai Jiao Tong University
School of Life Science and Biotechnology
State Key Laboratory of Microbial Metabolism
Dongchuan Road 800
Shanghai, Shanghai 200240
China

Re: Spectrum03011-22R2 (Phosphorylation of the yeast lysine methyltransferase Set5 is associated with the differential expression and phosphorylation of the mitogen-activated protein kinase Hog1 under prolonged acetic acid stress)

Dear Prof. Xin-Qing Zhao:

Link Not Available

Sincerely,

Jing Han

Journals Department
Reviewer comments:

Reviewer #3 (Comments for the Author):

The authors have addressed all of my concerns.

Reviewer #4 (Comments for the Author):

Some of my concerns with this manuscript have been addressed by the authors.

I continue to have reservation around the use of the Set5 antibody. The data does show a strong enrichment for the HOG1 coding sequence, which is the main point the authors wish to make but it would be nice to know that Set5 was the only thing being pulled down and that the mutation of serine and threonine residues is not altering Set5 stability or abundance as this could alter the ChIP result.

The authors have elected not to change the labelling of figure 4B and 4D. The figure is labelled "Relatively changed abundance". Based on what is described, western blots have been performed in triplicate for Hog1 or Hog1-P in SPSC01 and PLY01 and strains harboring the SET5 phosphosite mutants. The ECL signals were integrated and mean values determined. In 4B the SPSC01 mean is normalized to 1.00 and PLY01 values are compared to the SPSC01 value. In 4D the PLY01 value is normalized to 1.00 and the value of Hog1 and Hog1-P signals are displayed relative to PLY01. Thus, it appears to me that the graphs display relative abundance, not relative change in abundance. If I am incorrect here then the authors need to clarify how they have normalized and presented this data since relative abundance and relative change in abundance are not the same thing.

The manuscript still needs editing for English language as there are unclear statements for example Line 47-49: "... the prevention of fungi that are harmful for bioproduction and food preservation." are you saying prevention of contamination by fungi, or the prevention of fungal contamination?

Staff Comments:

Preparing Revision Guidelines

Please return the manuscript within 60 days; if you cannot complete the modification within this time period, please contact me. If you do not wish to modify the manuscript and prefer to submit it to another journal, please notify me of your decision immediately so that the manuscript may be formally withdrawn from consideration by Microbiology Spectrum.

Point-to-point response to reviewers

Some of my concerns with this manuscript have been addressed by the authors.

I continue to have reservation around the use of the Set5 antibody. The data does show a strong enrichment for the *HOG1* coding sequence, which is the main point the authors wish to make but it would be nice to know that Set5 was the only thing being pulled down and that the mutation of serine and threonine residues is not altering Set5 stability or abundance as this could alter the ChIP result.

Response from the authors: We appreciate the thoughtful comment from the reviewer, and more delicate experiments will be performed in our future studies. It will be nice to reveal whether the mutations of serine and threonine residues affect Set5 stability or abundance if we have the suitable Set5 antibody. As we have explained in the previous review response, the method we used is classical to study the effects of protein phosphorylation on regulation of gene expression (EMBO J. 2022, 41(4): e108290; J Biol Chem. 2021, 296: 100164). So far, no negative effects of the mutations on stability or abundance of the mutant proteins have been reported. However, we agree with the reviewer that it is valuable to verify the mutation effects in further studies. We have added related discussion in the manuscript (Line 310-314).

The authors have elected not to change the labelling of figure 4B and 4D. The figure is labelled "Relatively changed abundance". Based on what is described, western blots have been performed in triplicate for Hog1 or Hog1-P in SPSC01 and PLY01 and strains harboring the SET5 phosphosite mutants. The ECL signals were integrated and mean values determined. In 4B the SPSC01 mean is normalized to 1.00 and PLY01 values are compared to the SPSC01 value. In 4D the PLY01 value is normalized to 1.00 and the value of Hog1 and Hog1-P signals are displayed relative to PLY01. Thus, it appears to me that the graphs display relative abundance, not relative change in abundance. If I am incorrect here then the authors need to clarify how they have normalized and presented this data since relative abundance and relative change in abundance are not the same thing.

Response from the authors: We appreciate the comment from the reviewer, and have changed the description to "Relative abundance" in the revised figure.

The manuscript still needs editing for English language as there are unclear statements for example Line 47-49: "... the prevention of fungi that are harmful for bioproduction and food preservation." are you saying prevention of contamination by fungi, or the prevention of fungal contamination?

Response from the authors: We thank the reviewer for the comment. We would like to

state to prevent growth of harmful fungi. We have modified the sentence (Line 48). We have also thoroughly checked and improved the English writing, and the revision was marked in red in the revised manuscript.

February 27, 2023

Prof. Xin-Qing Zhao
Shanghai Jiao Tong University
School of Life Science and Biotechnology
State Key Laboratory of Microbial Metabolism
Dongchuan Road 800
Shanghai, Shanghai 200240
China

Re: Spectrum03011-22R3 (Modification of phosphorylation sites in the yeast lysine methyltransferase Set5 exerts influences on the mitogen-activated protein kinase Hog1 under prolonged acetic acid stress)

Dear Prof. Xin-Qing Zhao:

Your manuscript has been accepted, and I am forwarding it to the ASM Journals Department for publication. You will be notified when your proofs are ready to be viewed.

Sincerely,

Jing Han
Editor, Microbiology Spectrum
